# Human-Specific Regulation of Neurotrophic Factors MANF and CDNF by microRNAs

**DOI:** 10.3390/ijms22189691

**Published:** 2021-09-07

**Authors:** Julia Konovalova, Dmytro Gerasymchuk, Sergio Navarette Arroyo, Sven Kluske, Francesca Mastroianni, Alba Vargas Pereyra, Andrii Domanskyi

**Affiliations:** 1Institute of Biotechnology, HiLIFE, University of Helsinki, Viikinkaari 5D, 00790 Helsinki, Finland; julia.konovalova@helsinki.fi (J.K.); dmytro.gerasymchuk@helsinki.fi (D.G.); serginaryo@gmail.com (S.N.A.); svenk1998@gmx.de (S.K.); francesca.mastroianni@studenti.units.it (F.M.); alba.pereyra.20@ucl.ac.uk (A.V.P.); 2Institute of Molecular Biology and Genetics, NASU, 03143 Kyiv, Ukraine

**Keywords:** mesencephalic astrocyte derived neurotrophic factor, MANF, cerebral dopamine neurotrophic factor, CDNF, microRNAs, 3′UTR, human-specific regulation

## Abstract

Mesencephalic astrocyte derived neurotrophic factor (MANF) and cerebral dopamine neurotrophic factor (CDNF) are novel evolutionary conserved trophic factors, which exhibit cytoprotective activity via negative regulation of unfolded protein response (UPR) and inflammation. Despite multiple reports demonstrating detrimental effect of MANF/CDNF downregulation, little is known about the control of their expression. miRNAs—small non-coding RNAs—are important regulators of gene expression. Their dysregulation was demonstrated in multiple pathological processes and their ability to modulate levels of other neurotrophic factors, glial cell line-derived neurotrophic factor (GDNF) and brain-derived neurotrophic factor (BDNF), was previously reported. Here, for the first time we demonstrated direct regulation of MANF and CDNF by miRNAs. Using bioinformatic tools, reporter assay and analysis of endogenous MANF and CDNF, we identified that miR-144 controls MANF expression, and miR-134 and miR-141 downregulate CDNF levels. We also demonstrated that this effect is human-specific and is executed via predicted binding sites of corresponding miRNAs. Finally, we found that miR-382 suppressed hCDNF expression indirectly. In conclusion, we demonstrate for the first time direct regulation of MANF and CDNF expression by specific miRNAs, despite the fact their binding sites are not strongly evolutionary conserved. Furthermore, we demonstrate a functional effect of miR-144 mediated regulation of MANF on ER stress response markers. These findings emphasize that (1) prediction of miRNA targets based on evolutionary conservation may miss biologically meaningful regulatory pairs; and (2) interpretation of miRNA regulatory effects in animal models should be cautiously validated.

## 1. Introduction

Mesencephalic astrocyte derived neurotrophic factor (MANF) and cerebral dopamine neurotrophic factor (CDNF) represent a family of novel, evolutionary conserved proteins [1]. MANF and CDNF reside in the endoplasmic reticulum (ER) lumen and are involved in ER homeostasis and stress response [2]. They are classified as separate family of neurotrophic factors and are extensively studied due to their protective activity [3]. Their trophic effect was demonstrated in multiple models of neurodegenerative diseases, such as Parkinson’s disease (PD), Alzheimer’s disease (AD), retinal disorders, etc. (reviewed in [2]). Although MANF and CDNF have been discovered and named after their neurotrophic activity, they show no structural resemblance to other neurotrophic factors, such as brain-derived neurotrophic factor (BDNF) or glial cell line-derived neurotrophic factor (GDNF), have no currently identified cell surface receptors, and both are also expressed and act outside of the central nervous system (CNS), including heart, liver, pancreas, kidney and bones, and have been shown to implicate in diseases such as diabetes and chondrodysplasia [2,4]. Their great therapeutic potential has led to ongoing phase 1/2 clinical trial of CDNF in Parkinson’s disease (ClinicalTrials.gov: NCT03295786).

microRNAs (miRNAs) are short non-coding RNAs, estimated to control the expression of up to 60% protein-coding genome by guiding multiprotein RNA-induced silencing complex to their target mRNAs [5,6]. miRNAs regulate gene expression usually by destabilizing target mRNAs or inhibiting their translation, which is commonly achieved by binding to 6–8 nt long seed region on 3′ untranslated regions (3′UTRs) of the target [6]. miRNAs have been shown to participate in multiple physiological processes, such as cell proliferation and differentiation, response to oxidative stress, endocytosis and cell motility [7,8,9]. Disruption in miRNA biogenesis is extensively studied in various pathological conditions, including neurodegenerative disorders [9,10,11,12,13], diabetes [14] and liver diseases [15].

One of possible ways miRNAs can implicate in neurodegeneration is via regulation of neurotrophic factors expression. Several studies demonstrate that miRNAs can control and fine-tune the expression of neurotrophic factors, such as GDNF and BDNF [16,17,18]. However, despite great interest in studying MANF and CDNF due to their therapeutic potential, not much is known about their regulation on post-transcriptional level. Several studies have reported alterations in expression of MANF or CDNF in the context of altered miRNA expression profile [19,20,21,22]. However, the ability of miRNAs to directly regulate endogenous MANF and CDNF remained unaddressed.

The goal of current study was to elucidate ability of selected miRNAs to directly regulate the expression of human MANF (hMANF) and CDNF (hCDNF). We assessed regulation of hMANF protein expression by miR-141, miR-144, miR-544a and miR-338, and tested the effect of miR-134, miR-141, miR-190a, miR-382, miR-539 and miR-599 on hCDNF protein levels. Using bioinformatic tools, reporter assay and analysis of endogenous MANF and CDNF (Figure 1), our study demonstrates that miR-144, and miR-134 and miR-141 are direct regulators of hMANF and hCDNF expression, respectively. Additionally, we observed indirect inhibitory effect of miR-382 on hCDNF. Importantly, the effect of identified miRNAs on hMANF and hCDNF is human-specific: no inhibition of reporter protein levels has been detected with mouse MANF and CDNF 3′UTR constructs. Therefore, for the first time we identified miRNAs, capable to regulate the levels of endogenous hMANF and hCDNF. In addition, we demonstrated that despite miRNA binding cites in the 3′UTRs of hMANF and hCDNF are not strongly evolutionary conserved, they are nevertheless regulated by miRNAs. These data imply that miRNA target prediction algorithms based on evolutionary conservation of binding sites may be missing biologically meaningful regulatory pairs. Current findings also emphasize the necessity of using human models in studying biological effect of miRNAs.

## 2. Results

### 2.1. miRNAs, Predicted to Bind 3′UTR of Human MANF and CDNF

In order to identify miRNA candidates targeting hMANF and hCDNF 3′UTR, in silico analysis using publicly available web-based algorithm miRanda was used as a main source [23]. According to this algorithm, 3′UTR of hMANF contains seed sequences for miRNAs miR-144, miR-544a and miR-338, and 3′UTR of hCDNF contains seed sequences for miR-134, miR-190a, miR-382, miR-539 and miR-599 (Figure 2).

Some of these miRNAs were also predicted to target 3′UTRs of hMANF and hCDNF by other bioinformatic tools, such as TargetScan [24] (as poorly conserved) and microT-CDS (Diana Tools) [25,26] (Appendix A). For example, all miRNA candidates predicted to target hCDNF were also identified by both TargetScan and microT-CDS (Diana Tools). In case of hMANF, miR-338 was also predicted by TargetScan.

While studying miRNAs, predicted by microT-CDS (Diana Tools) to target hMANF and hCDNF 3′UTRs, we noticed that there are putative seed sequences for miR-141 on 3′UTRs of both hMANF and hCDNF. Therefore, this miRNA was also added to the list of candidates for further analysis.

### 2.2. miRNA Regulation of Protein Synthesis from Transcripts Containing hMANF and hCDNF 3′UTRs

In order to confirm that predicted miRNAs are capable to regulate the expression of corresponding protein via 3′UTR of hMANF or hCDNF, we utilized dual-luciferase reporter assay (Figure 3A). 3′UTR of hMANF or hCDNF was cloned downstream of Renilla luciferase coding sequence. The same construct has Firefly luciferase coding sequence, which expression is independent of Renilla and used as transfection efficiency control. The obtained plasmids were co-transfected with mimics (Appendix A) of miRNA candidates into HEK293T cells. Scrambled miRNA was used as a negative control (Scrb). 48 h later cells were lysed, and luciferase activity was measured.

We showed that miR-144 and miR-544a inhibit the expression of Renilla luciferase under regulation of hMANF-3′UTR (Figure 3B). Interestingly, the only conserved miRNA—miR-338—showed no regulatory effect on reporter protein levels. No significant effect in the same reporter assay was observed for miR-141 either. Testing the ability of miRNAs to regulate protein synthesis containing hCDNF 3′UTR transcript demonstrated that poorly conserved miRNAs miR-134, miR-141, miR-190a and miR-382 significantly inhibited the expression of Renilla luciferase (Figure 3C). In contrast, there was no downregulation of the luciferase activity after treatment with miR-539 and miR-599.

To test the specificity of the regulation of hMANF and hCDNF, we assessed if selected miRNA candidates can regulate protein synthesis via 3′UTRs of mouse (m) MANF and CDNF (Appendix A). For this experiment, 3′UTR sequences of mMANF and mCDNF were cloned downstream of Renilla luciferase coding sequence and dual-luciferase assay was performed. None of the selected miRNAs (including conserved miR-338) showed inhibition of luciferase signal from transcripts containing both mMANF and mCDNF 3′UTRs (Figure 3D,E).

### 2.3. Regulation of Endogenous hMANF and hCDNF by miRNAs

As the next step, we aimed to confirm if miRNAs, regulating protein synthesis in a reporter assay with hMANF and hCDNF 3′UTRs, can also regulate endogenous hMANF and hCDNF. HEK293T cells, containing detectable amount of MANF and CDNF, were transfected with miR-144 and miR-544a for the analysis of hMANF regulation, or miR-134, miR-141, miR-190a and miR-382 for the analysis of hCDNF regulation. Corresponding siRNAs were used as positive controls. Levels of mRNA for either MANF or CDNF were measured by quantitative PCR (qPCR). miR-144, but not miR-544a, was able to significantly downregulate hMANF mRNA expression (Figure 4A). In hCDNF analysis, no miRNAs significantly decreased mRNA levels (Figure 4B), however, there was a trend for hCDNF mRNA downregulation by miR-134 (*p*-value = 0.06) with large effect size (η^2^_p_ = 0.860).

As miRNAs can induce translational silencing, but not necessarily degradation, of their mRNA targets, we also analyzed the protein levels of hMANF and hCDNF in cell lysates by corresponding ELISAs. In accordance with qPCR data, miR-144 but not miR-544a showed statistically significant downregulation of hMANF protein levels (Figure 4C). For hCDNF, we found that miR-134, miR-141 and miR-382 suppressed hCDNF protein synthesis, while miR-190a had no significant effect on hCDNF protein expression (Figure 4D).

### 2.4. Confirmation of miRNAs’ Predicted Binding Sites

Identification of miRNAs affecting protein synthesis does not distinguish between their direct and indirect regulation. Therefore, to assess if validated miRNAs carry on regulation via predicted binding sites, we introduced specific point mutations within predicted seed sequences (Figure 5A).

Mutation of predicted miR-144 binding site fully abolished the ability of miR-144 to inhibit reporter protein synthesis, confirming that miR-144 acts via the predicted binding site (Figure 5B). Interestingly, independently of miR-144, the mutation also resulted in decreased luciferase signal in comparison to the construct with wild type (wt) hMANF 3′UTR. Predicted binding site was also confirmed for miR-134 (Figure 5C): mutation of its corresponding seed sequences fully eliminated its inhibitory effect on protein synthesis. In contrast, mutation of miR-141 binding site reduced, but did not fully block, its effect on protein synthesis (Figure 5D). In agreement with this data, microT-CDS (Diana Tools) predicts the second binding site for miR-141. Therefore, the existence of this additional binding site can be one of the possible explanations for this result. In contrast, mutation of predicted binding site had no effect of inhibitory function of miR-382 (Figure 5E), indicating that this miRNA regulates hCDNF indirectly or via unknown binding site.

### 2.5. Physiological Effect of miRNAs, Targeting Expression of Human MANF and CDNF

As it was discussed previously, MANF and CDNF play important role in the regulation of ER stress response. Therefore, to test if the treatment with selected miRNAs interferes with hMANF and hCDNF function, HEK293T cells were treated with miRNAs and expression of selected ER stress markers were measured with qPCR. Transfection with miRNAs or siRNAs, targeting hMANF or hCDNF, had no statistically significant effect on expression of selected ER stress markers under non-stress conditions (Figure 6B–E). Therefore, mild stress was induced using tunicamycin (TM; 2 µg/mL for 24 h)—a well-known ER stress inducer [27] (Figure 6A). qPCR results demonstrated that treatment of cells with miR-144 decreased expression of XBP1s (Figure 6E) and had a trend to reduce levels of ATF6 and CHOP (Figure 6C,D). This effect was similar to the one observed in the cells treated with siRNA targeting hMANF (siMANF), where significant downregulation of ATF6 was detected (Figure 6C). However, no effect on expression of ER stress markers was found in the cells treated with miRNAs and siRNA targeting hCDNF (siCDNF). Only miR-134 significantly increased the expression of BiP mRNA (Figure 6B), but this effect was not observed in the cells treated with siCDNF.

## 3. Discussion

In this study, we aimed to identify miRNAs regulating neurotrophic factors CDNF and MANF in human cells. MANF and CDNF play important role regulating response to ER stress, which is a common hallmark of multiple degenerative diseases. ER stress triggers the unfolded protein response (UPR), which initially plays protective role. However, when prolonged, UPR signaling becomes pro-apoptotic [28]. Multiple studies show that MANF and CDNF are negative regulators of the UPR and their deficiency is causing deteriorating effect in different cell types. For example, MANF-deficiency leads to chronic UPR activation in mouse brain [29] and impairment of neurites outgrowth in developing mouse cortex [30]. MANF knockdown in SH-SY5Y cells also increases toxicity in response to β-amyloids treatment [31]. In addition, MANF knockout activates all three branches of the UPR and increases apoptosis of pancreatic β-cells [32], and most recent report demonstrated that knockout of MANF gene induced mild ER stress in human embryonic stem cells-derived β-cells [4]. CDNF was also shown to regulate the UPR, activated in response to β-amyloids in hippocampal neurons [33]. Furthermore, CDNF knockout in mice results in age-related neurodegeneration of enteric neurons and altered function of dopamine system, observed in early stage of PD [34].

There are few reports demonstrating alterations in MANF and CDNF in humans. Mutated *MANF* gene was reported in patients with neurological abnormalities and diabetes [4,35]. There are several reports, showing alterations in levels of circulating MANF/CDNF in patients with diabetes [36], autoimmune diseases (rheumatoid arthritis and systemic lupus erythematosus) [37], PD [38] and stroke [39]. In addition, elevated levels of CDNF, but not MANF, were reported in the hippocampi of PD patients [40]. Altered miRNA profiling were reported in these diseases.

Dysregulation of cellular miRNA networks have been observed in aging and stress conditions. For example, predominant downregulation of miRNAs was reported in brains of aged mice [41], in motor neurons of ALS (amyotrophic lateral sclerosis) patients and in cells, exposed to ER and oxidative stress [10]. In addition, our previous study [42] showed that there is downregulation of Dicer expression in dopamine neurons of aged mice, accompanied by general decrease in miRNAs expression. There are also data demonstrating reduced Dicer levels in dopamine neurons of PD patients [43,44], as well as alterations in regulatory miRNA-mRNA networks in PD [45]. Abnormalities in miRNA networks were also reported in the context of various pathological conditions outside of CNS, such as in pancreas and liver (reviewed [14,15], respectively).

To regulate its targets, miRNAs usually bind to corresponding sites in 3′UTRs, though in some cases binding to other mRNA regions can also occur [7]. Binding of miRNA to its target is affected by multiple factors and is challenging to predict using computational methods. One of the most frequently used strategies to predict miRNAs, which may regulate specific target is based on the evolutionary conservation of miRNA binding sites in its 3′UTR; this is implemented in TargetScan prediction algorithm [24]. Interestingly, despite human CDNF 3′UTR being relatively large (3401 nt long), TargetScan did not predict any broadly conserved miRNA binding sites in its sequence. For human MANF 3′UTR with a length of 717 nt, TargetScan found only one conserved binding site for miR-338, however, also with lower probability of preferential conservation. Considering relatively widespread involvement of miRNAs in translational regulation, with miRNAs predicted to regulate around half of all protein coding mRNAs [46], we hypothesized that, despite the absence of broad evolutionary conservation in 3′UTR sequences of hCDNF and hMANF, miRNAs may still regulate them in a species-specific manner. Indeed, there are several reports showing dysregulation of either MANF or CDNF in condition of altered miRNAs expression (e.g., in case of disease or experimental treatments) [19,20,21,22]. However, in contrast to other neurotrophic factors, such as BDNF and GDNF [16,17,18], currently there are no studies confirming direct regulation of MANF or CDNF by miRNAs.

Using miRanda and microT-CDS algorithms [23,25,26], we identified nine miRNA binding sites (miR-134, miR-141, miR-144, miR-190a, miR-338, miR-382, miR-539, miR-544a and miR-599) in hMANF and hCDNF 3′UTRs, which were also among the list of poorly conserved sites predicted by TargetScan. We then studied the ability of these nine miRNAs to regulate the expression of MANF and CDNF in HEK293T cells. Unlike previous reports, we demonstrated the ability of selected candidates to directly regulate the expression of MANF and CDNF. Furthermore, we showed that regulation of MANF by miR-144, and CDNF by miR-134 and miR-141 is human-specific and confirmed their binding sites in corresponding 3′UTRs with site-directed mutagenesis. In addition, we report indirect regulation of hCDNF protein by miR-382. With current data we cannot fully exclude the possibility that identified miRNAs can regulate the expression of mouse MANF and CDNF. However, with our dual-luciferase reporter assay results we can state that if this regulation occurs, it is not implemented via 3′UTRs.

Interestingly, miRNAs identified to regulate MANF and CDNF have been reported to be upregulated in patients, suffering from various neurological conditions. For example, increased levels of miR-144 have been identified in cingulate gyrus of PD patients and levels of miR-144 expression also correlated with disease stage [47]. Furthermore, miR-144 upregulation was detected in cortex of AD and spinocerebellar ataxia type 1 patients [48]. Increase of miR-134 levels was reported in the brain of AD patients [49], and miR-134 showed its validity as a potential biomarker in patients with mild cognitive impairment and AD [50,51]. In addition, miR-144 was also found to be upregulated in blood of type 2 diabetes male patients [52].

Many degenerative diseases, such as diabetes and PD, are characterized by elevated ER stress. miRNA expression was reported to regulate and be regulated by ER signaling (reviewed in [53]). ER stress modulates expression of many miRNAs, e.g., via suppressing their biogenesis by translocation of Ago and Dicer proteins in stress granules or reducing expression of these proteins [10,54]. However, there is also a fraction of miRNAs, whose expression is upregulated under stress conditions [41,42]. In addition, multiple miRNAs were reported to affect expression of ER signaling members [53]. These results suggest that when cells are struggling with ER stress, it affects miRNA biogenesis. Dysregulation in miRNAs expression further exacerbate ER stress, e.g., by affecting levels of UPR-involved proteins, making cells more vulnerable to additional stressors, and further affecting miRNAs biogenesis, forming vicious cycle eventually contributing to cell death.

The effect of miRNAs on the expression levels of proteins is usually mild [55]. However, it can be speculated that prolonged mild downregulation of MANF/CDNF may contribute to impairment of ER stress response, making cells more vulnerable for various insults and implicating in development and progression of various diseases, such as neurodegeneration [34] or diabetes [4]. In support of this hypothesis, here we demonstrated that treatment of HEK293T cells with miR-144 or siMANF affects expression of some ER stress markers under prolonged mild stress condition. However, to the best of our knowledge, there is no data on the molecular mechanisms regulating the expression of miRNAs studied in our manuscript. In fact, factors regulating expression levels of most miRNAs are not studied. Considering our results on the effects of miRNAs targeting MANF/CDNF, particularly miR-144, on ER stress markers, further studies of molecular pathways and factors regulating miR-144 in stress conditions are necessary. Interestingly, no effect of CDNF downregulation on ER stress response was observed in our study, in contrast to previous data [33,34]. These discrepancies can be explained by different models used.

Curiously, the effect of miR-144 on ER stress markers’ expression was more prominent than knockdown of MANF with siRNA. One miRNA can control expression of several genes, affecting cell function via multiple pathways. For example, miRNAs identified to regulate expression of human MANF/CDNF may be also implicated in increased oxidative stress in PD [9]. Increased miR-144 can cause impaired antioxidant response and UPR via downregulation of Nrf2 [56,57]. In addition, increased levels of miR-320a, which was reported to have negative correlation with hMANF mRNA expression [19], can contribute to accumulation of α-synuclein—main hallmark of PD [58]. Furthermore, miR-144 was shown to suppress migration and proliferation of microvascular epithelial cells [59], corresponding with data of MANF implication in cell migration [60] and proliferation [32], although in other cell types. Taking together, these data emphasize necessity to further study physiological effect of these miRNAs.

Both MANF and CDNF are widely expressed throughout the body and exhibit their protective functions in various types of tissues [2]. By now, no alternative 3′UTR sequences were reported for MANF/CDNF mRNA, therefore we suggest that the results obtained in this study could be applied to other cell types. However, molecular mechanism of miRNA action can be tissue specific. There are multiple reports suggesting that regulation of gene expression by miRNA can depend on multiple factors, such as endogenous expression of miRNAs, and number and level of expression of target genes [61,62,63]. Therefore, additional studies are required to further investigate physiological effect of selected miRNAs and contribution of MANF/CDNF in this effect in degenerative diseases. Antagomirs and/or specific target protectors can be valuable tools for these purposes with potential therapeutic use in the future [64,65].

Our study also demonstrated that selected miRNAs carry out MANF/CDNF control in human-specific way. Considering that MANF and CDNF are evolutionary conserved proteins, it raises a question about their human-specific functions. In addition, these findings bring to attention possible human-specific regulatory effect of miRNAs on various physiological processes of an organism. Furthermore, our results emphasize that translation of findings about regulation of gene expression by miRNAs from rodent models to humans should be done with extra precautions.

## 4. Materials and Methods

### 4.1. Bioinformatic Prediction of miRNAs, Targeting hMANF and hCDNF

In order to identify miRNAs, targeting MANF and CDNF, web-based predicting algorithm miRanda (http://www.microrna.org/microrna/home.do, accessed on 25 October 2016) was used as a main source of candidates. Additionally, Diana Tool microT-CDS (http://diana.imis.athena-innovation.gr/DianaTools/index.php?r=microT_CDS/index, accessed on 20 January 2017) and TargetScan (http://www.targetscan.org/vert_72/, accessed on 11 December 2020) were used.

### 4.2. Cell Culture

HEK293T (ATCC, CRL-1573) cells were maintained in Dulbecco’s Modified Eagle Medium (DMEM; D-7777, Sigma-Aldrich, Saint Louis, MO, USA), supplemented with 10% heat-inactivated foetal bovine serum (#10270-106, Gibco, Waltham, MA, USA) and normocin (100 µg/mL; Invitrogen, Waltham, MA, USA), under humidified conditions with 5% CO_2_ at 37 °C.

### 4.3. Plasmids

Human MANF and CDNF 3′UTR sequences were PCR amplified from human genomic DNA (G1471, Promega, Madison, WI, USA) using specific primers (Appendix A) with Phusion High-Fidelity DNA Polymerase (F530S, Thermo Fisher Scientific, Waltham, MA, USA) according to manufacturer’s protocol. The PCR products were digested with FastDigest XhoI (FD0694, Thermo Fisher Scientific, Waltham, MA, USA) and NotI (FD0595, Thermo Fisher Scientific, Waltham, MA, USA) restriction enzymes and cloned downstream of Renilla luciferase gene in psiCHECK™-2 vector (C8021, Promega, Madison, WI, USA), linearized with the same restriction enzymes. Obtained vectors psiCHECK-hMANF-3′UTR and psiCHECK-hCDNF-3′UTR were also used to generate sequences with mutations in predicted binding sites by site-directed mutagenesis. Correct cloning of all constructs was confirmed by sequencing.

3′UTR of mouse MANF and CDNF were PCR amplified from genomic DNA (mgDNA), isolated from wild-type female C57Bl/6N mouse. 3′UTR of mCDNF was amplified using primers listed in Appendix A. 3′UTR of mouse MANF was amplified in two steps. First, bigger fragment of MANF sequence was amplified from mgDNA using primers listed in Appendix A. Then the obtained fragment was used as a template for PCR amplification of mouse 3′UTR using a second set of primers (Appendix A). Amplified sequences were cloned in psiCHECK™-2 vector (C8021, Promega, Madison, WI, USA) downstream of Renilla luciferase, using an InFusion kit (638933, Takara Bio Inc., San Jose, CA, USA) according to manufacturer’s instructions.

### 4.4. Site-Directed Mutagenesis

Site directed mutagenesis of plasmids psiCHECK-hMANF-3′UTR and psiCHECK-hCDNF-3′UTR was performed with Phusion site-directed mutagenesis kit (F-541, Thermo Fisher Scientific, Waltham, MA, USA), according to the manufacturer’s protocol. PCR reaction was performed using psiCheck-hMANF3′UTR and psiCheck-hCDNF3′UTR plasmids as a template (primers sequences are in Appendix A). All constructs were verified by sequencing.

### 4.5. Dual-Luciferase Reporter Assay

To perform luciferase assay, HEK293T cells were seeded on 96-well plate (10,000 cells/well in 100 µL of cell culture medium). After incubation for 24 h, cells were transfected with corresponding plasmid (final concentration 1 ng/µL) and miRNA mimic (final concentration 50 nM; Appendix A) using DharmaFECT^TM^ Duo (0.1 µL/well, T-2010-02, Dharmacon, Lafayette, CO, USA) transfection reagent according to manufacturer’s protocol. 48 h after transfection, cells were lysed with 50 µL Passive Lysis Buffer per well. Lysates were used to measure luminescence with Dual-luciferase^®^ reporter assay system (Promega, E1910, Madison, WI, USA) according to the manufacturer’s protocol with VICTOR3 (Perkin Elmer, Waltham, MA, USA) or Varioskan™ LUX reader (Thermo Fisher Scientific, Waltham, MA, USA). Briefly, 50 µL of Luciferase Assay Reagent II were added to 10 µL of cell lysate to measure firefly luciferase (FL) activity, followed by 50 µL of Stop & Glo Reagent to measure Renilla luciferase (RL) activity. RL/FL ratio was used for statistical analysis. Each experiment was repeated 3–4 times with 3–4 wells per experiment.

To validate obtained results, HEK293T cells were transfected with plasmid with mutation in predicted miRNA binding site and corresponding mimic, and 48 h later they were harvested and assayed with Dual-lusiferase^®^ reporter assay system as described above.

### 4.6. Quantitative Real-Time PCR (RT-qPCR)

To analyze effect of validated miRNAs on expression of endogenous hMANF and hCDNF mRNA, HEK293T cells were seeded on 12-well plate (150,000 cells/well) and 24 h later transfected with mimics (final concentration 5 nM) using DharmaFECT™ transfection reagent (T-2001-02, Dharmacon, Lafayette, CO, USA) according to manufacturer’s protocol. siRNAs for hMANF (M-012158-03-0005, Dharmacon, Lafayette, CO, USA) and hCDNF (M-034281-01-0005, Dharmacon, Lafayette, CO, USA) were used as positive controls.

48 h post-transfection, cells were washed with PBS, lysed with TRI Reagent™ Solution (AM9738, Invitrogen, Waltham, MA, USA) and homogenized on the plate. To extract RNA, chloroform (32211, Sigma-Aldrich, Saint Louis, MO, USA) was added to cell lysate. Mixture was collected, shaken vigorously, incubated for 2–3 min at RT and then centrifuged for 15 min at 12,000× *g* at 4 °C. Then aqueous phase was transferred to the fresh tube, mixed with isopropanol (59300, Sigma-Aldrich, Saint Louis, MO, USA), incubated for 10 min at RT and centrifuged for 10 min at 12,000× *g* at 4 °C. After centrifugation, RNA pellet was washed with 75% ethanol and centrifuged for 5 min at 7500× *g* at 4 °C. Then ethanol was removed, pellet was left to air dry for approximately 5 min and resuspended in 10–20 µL of RNase-free water. RNA concentration was measured using NanoDrop ND 1000 Spectrophotometer (Thermo Fisher Scientific, Waltham, MA, USA) and NanoDrop 1000 3.8.1 software. RNA was stored at −80 °C.

cDNA was synthesized using 500–2000 ng of RNA (equal amount of RNA was used withing the same experiment) using Maxima H Minus Reverse Transcriptase kit (#EP0753 Thermo Fisher Scientific, Waltham, MA, USA) according to the manufacturer’s instructions. Briefly, 1 µL of Oligo(dT)18 (100 pmol, SO132, Thermo Fisher Scientific, Waltham, MA, USA) and 1 µL of dNTP mix (10 nM, Thermo Fisher Scientific, Waltham, MA, USA) were mixed with 13 µL of RNA sample (equalized to the same amount with RNase-free water) and incubated for 5 min at 65 °C. Then 5 µL of mix, containing 5× RT buffer and Maxima H Minus Reverse Transcriptase (200U, EP0753, Thermo Fisher Scientific, Waltham, MA, USA), was added, gently mixed, briefly centrifuged and incubated for 30 min at 50 °C and then for 5 min at 85 °C. No RNA template control was included in each experiment. cDNA was cooled on ice, diluted 1:5, and stored at −20 or used immediately for RT-qPCR.

RT-qPCR was performed using LightCycler^®^ 480 System (Roche Molecular Diagnostics, Basel, Switzerland) using the LightCycler^®^ 480 Software release 1.5.1.62 in a final volume of 10 µL on 384-well plate, sealed with adhesive plate sealer (04729749001, Roche Molecular Systems, Pleasanton, CA, USA). Each reaction included 2 µL of cDNA template, 2.5 µL of nuclease-free water, 5 µL 2XTaqMan™ Universal PCR Master Mix (4324018, Thermo Fisher Scientific, Waltham, MA, USA) and 0.5 µL TaqMan Gene expression primer: hMANF (Hs00180640_m1), hCDNF (Hs00418490_m1) and hGAPDH (Hs02758991_g1) as reference gene. Each reaction was run in duplicates with 2–3 replicates per treatment group in each experiment. Following qPCR program was used (1) pre-incubation: 50 °C for 2 min and 95 °C for 10 min; (2) amplification and data acquisition: 40 cycles of 95 °C for 15 s and 60 °C for 1 min; (3) cooling: 40 °C for 10 s. Data were analyzed according to the ΔΔCt method. Data were discarded if Ct values were ≥35 or Ct values between technical replicates >0.5.

To test the effect of miRNAs on the expression of ER stress markers, HEK293T cells were seeded on 24- (70,000cells/well) or 12-well plate (150,000 cells/well) and 24 h later transfected with mimics (final concentration 5nM) as described above. 48 h later cells were treated with 2 µg/mL tunicamycin (ab120296, Abcam, Cambridge, UK) or the same volume of DMSO (D2650-100ML, Sigma-Aldrich, Saint Louis, MO, USA). 24 h after treatment with tunicamycin, cells were lysed with TRI Reagent™ Solution (AM9738, Invitrogen, Waltham, MA, USA), and RNA was isolated as described above. cDNA was synthesized using 500–1000 ng of RNA (equal amount of RNA was used withing the same experiment) using Maxima H Minus Reverse Transcriptase kit (#EP0753, Thermo Fisher Scientific, Waltham, MA, USA) according to the manufacturer’s instructions, as described above. RT-qPCR was performed using LightCycler^®^ 480 System (Roche Molecular Diagnostics, Basel, Switzerland) as described above. Each reaction included 5 µL of LightCycler^®^ 480 SYBR^®^ Green I Master, 2x concentrated (04887352001, Roche Molecular Systems, Pleasanton, CA, USA), 2.1 µL of PCR grade water (Roche Molecular Systems, Pleasanton, CA, USA), 0.2 µL of 10 µM forward and 0.2 µL of 10 µM reverse primers (Appendix A), and 2.5 µL of cDNA template. Each reaction was run in duplicates. Following qPCR program was used: (1) pre-incubation: 95 °C for 5 min; (2) amplification and data acquisition: 45 cycles of 95 °C for 10 s, 60 °C for 10 s and 72 °C for 10 s; (3) melting curve: 95 °C for 5 s, 65 °C for 1 min and continuous slow increase in temperature until 97 °C; (4) cooling: 40 °C for 10 s. Data were analyzed according to the ΔΔCt method. Data were discarded if Ct values were ≥35 or Ct values between technical replicates > 0.5.

### 4.7. Enzyme-Linked Immunosorbent Assay (ELISA)

To assess effect of validated miRNAs on expression of endogenous hMANF and hCDNF protein, HEK293T cells were seeded on 24-well plate (70,000 cells/well) and 24 h later transfected with mimics (final concentration 5 nM) using DharmaFECT™ transfection reagent (T-2001-02, Dharmacon, Lafayette, CO, USA) according to manufacturer’s protocol. siRNAs for hMANF (M-012158-03-0005, Dharmacon, Lafayette, CO, USA) and hCDNF (M-034281-01-0005, Dharmacon, Lafayette, CO, USA) were used as positive controls.

72 h after transfection, cells were proceeded as described elsewhere [66]. Briefly, medium was removed and cells were washed with cold PBS, containing 0.5 mM Na_3_VO_4_, and incubated for 30 min on ice in cold-lysis buffer (137 mM NaCl, 20 mM Tris-HCl, 2.5 mM EDTA, 1% NP40, 10% glycerol, 0.5 mM Na_3_VO_4_ and protease inhibitor cocktail (Roche, 04693159001, Basel, Switzerland) on shaker. Lysates then were collected into fresh tubes and centrifuged at 12,000 rpm for 20 min at +4 °C. Supernatants were collected and either used immediately for analysis or stored at −80 °C.

Human CDNF was tested with CDNF ELISA, developed by Galli [38]. Briefly, MaxiSorp 96-well plate (Nunc, Thermo Fisher Scientific, Waltham, MA, USA) was coated with goat anti-hCDNF pAb (1 ng/mL; AF5097, R&D Systems, Minneapolis, MN, USA) in carbonate coating buffer (35 mM NaHCO_3_, 15 mM Na_2_CO_3_, pH 9.6) overnight at +4 °C. Next day, plate was washed with PBS/0.05% Tween 20 (PBST) and blocked with 3% BSA (A9647, Sigma-Aldrich, Saint Louis, MO, USA) in PBS for 2 h at RT. Then the plate was washed again with PBST and diluted in BSA standards of recombinant human CDNF (P-100-100, Icosagen, Tartu, Estonia) and experimental samples were added to the plate in duplicates and incubated overnight at +4 °C on shaker. Then, after repeated four times washings with PBST, plate was incubated with rabbit anti-hCDNF detection antibodies (100 ng/mL, produced in the Saarma lab, DDV1 [1]) diluted in BSA for 3 h at RT on shaker. After additional 4 washes with PBST, plate was incubated with HRP-linked donkey anti-rabbit antibodies (NA9340V, GE Healthcare, Chicago, IL, USA) in BSA for 2 h at RT on shaker. After last wash with PBST for 4 times, detection was performed using DuoSet ELISA Development System (DY999, R&D Systems, Minneapolis, MN, USA), according to manufacturer’s protocol. The absorbance was read by plate reader (VICTOR3, Perkin Elmer, Waltham, MA, USA) at 450 nm and 540 nm.

To examine hMANF levels after treatment, human MANF ELISA test described elsewhere [67] was used. Briefly, MaxiSorp 96-well plate (Nunc, Thermo Fisher Scientific, Waltham, MA, USA) was coated overnight at +4 °C with goat anti-hMANF pAb (1 µg/mL; AF3748, R&D Systems, Minneapolis, MN, USA) in carbonate coating buffer (35 mM NaHCO_3_, 15 mM Na_2_CO_3_, pH 9.6). Next day, plate was washed once with PBST and incubated with 1% casein (C8654, Sigma-Aldrich, Saint Louis, MO, USA) in PBST blocking buffer at RT for 2 h. After washing plate with PBST, diluted in blocking buffer standards with recombinant human MANF (P-101-100, Icosagen, Tartu, Estonia) and study samples were added to the plate in duplicates and incubated overnight at +4 °C on shaker. Then, after washing four time with PBST, plate was incubated with HRP-linked mouse anti-hMANF detection antibodies (1 µg/mL; 4E12, Icosagen, Tartu, Estonia) for 5 h at RT on a shaker. For detection DuoSet ELISA Development System (DY999, R&D Systems, Minneapolis, MN, USA) was used after washing plate with PBST 4 times. Plate reader (VICTOR3, Perkin Elmer, Waltham, MA, USA) was used to read absorbance at 450 nm and 540 nm.

MANF and CDNF levels were normalized to total protein concentrations, measured with DC Protein Assay kit (5000114, Bio-Rad, Hercules, CA, USA) according to manufacturer’s instructions. Absorbance was measured with Plate reader (VICTOR3, Perkin Elmer, Waltham, MA, USA) at 750 nm.

### 4.8. Statistical Analysis

Statistical analysis was performed using One-way ANOVA (for dual-luciferase reporter assay) or Repeated-measurement one- or two-way ANOVA (for qPCR and ELISA), using post-hoc test (Turkey’s or Dunette’s multiple comparison test). Mixed-effect analysis was performed in case of missing values. Statistical significance was defined as *p*-value < 0.05. Statistical tests were performed with GraphPad Prism software (GraphPad Software, Inc., San Diego, CA, USA). Data represented as mean ± SEM [68].

## 5. Conclusions

We identified for the first time several miRNAs, directly regulating the expression of neurotrophic factors MANF and CDNF and affecting their function under stress conditions in human cells. Identified miRNAs have been previously reported to be dysregulated in several pathological conditions, such as Parkinson’s disease and diabetes. We also demonstrated that this regulation is human specific, showing the necessity to validate pathology-associated miRNA findings from animal models in human cells.

## Figures and Tables

**Figure 1 ijms-22-09691-f001:**
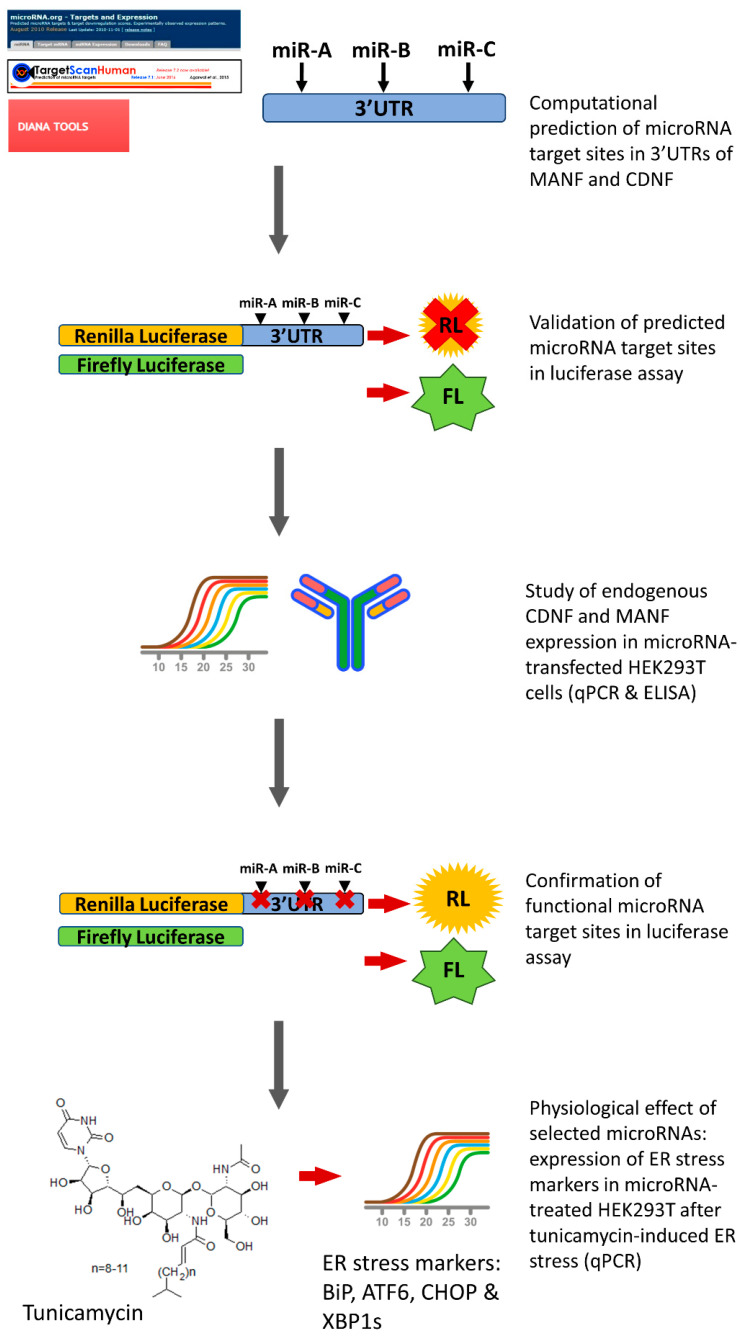
Experimental workflow of the study. Ability of miRNAs to regulate expression of MANF and CDNF was tested using bioinformatic tools, reporter assay and analysis of endogenous protein with qPCR and ELISA. In addition, physiological effect of selected miRNAs was tested with qPCR.

**Figure 2 ijms-22-09691-f002:**
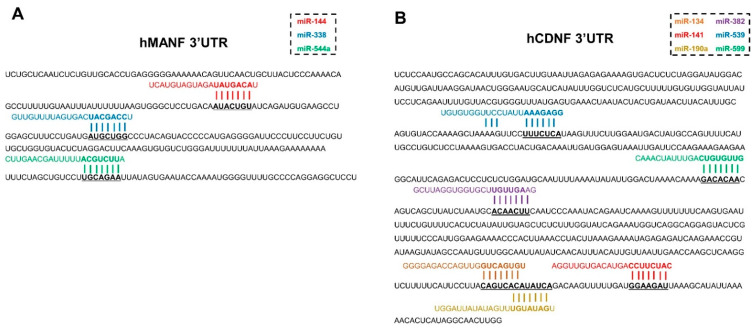
miRNAs predicted to target 3′UTRs of (**A**) hMANF and (**B**) hCDNF. Only miRNAs, included in the current study, are shown. Alignment is adapted from TargetScan and miRanda.

**Figure 3 ijms-22-09691-f003:**
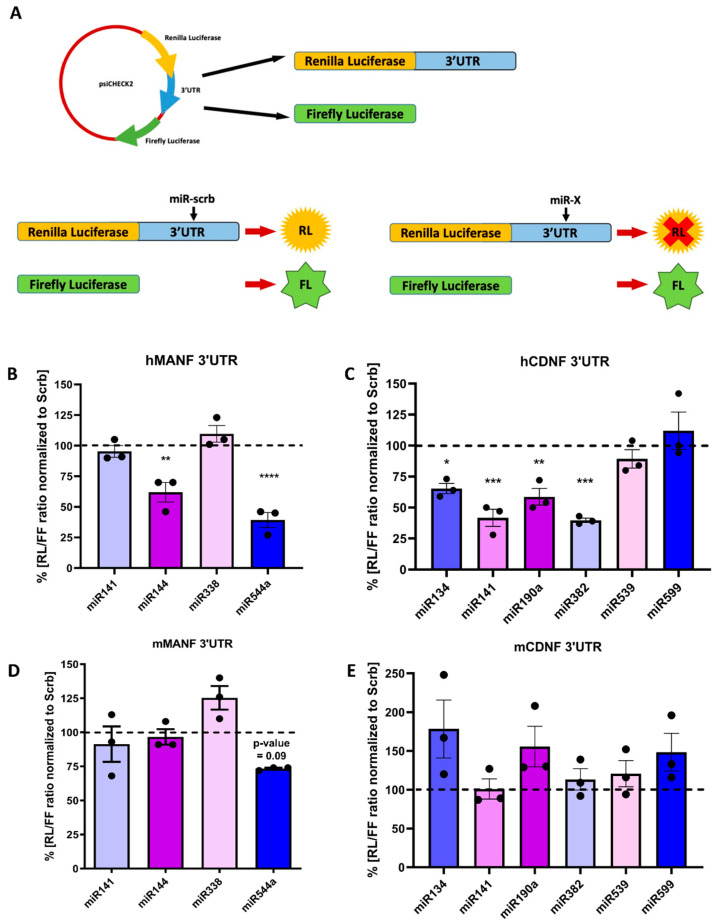
(**A**) Dual-luciferase reporter assay scheme: expression of Renilla luciferase is controlled by 3′UTR of gene of interest, while Firefly luciferase expression is independent and used as transfection efficiency control. Luciferase activity of reporter constructs, containing 3′UTRs of (**B**) human MANF, (**C**) human CDNF, (**D**) mouse MANF and (**E**) mouse CDNF, co-transfected with miRNA mimics. Each dot represents mean value from one independent experimental plate with 3–4 wells per group. *—*p*-value < 0.05, **—*p*-value < 0.01, ***—*p*-value < 0.001, ****—*p*-value < 0.0001.

**Figure 4 ijms-22-09691-f004:**
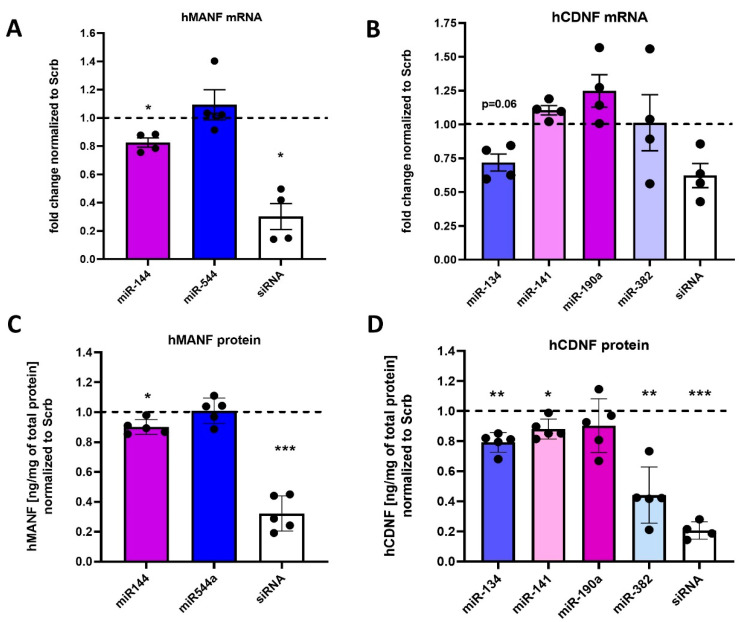
Regulation of endogenous hMANF (**A**,**C**) and hCDNF (**B**,**D**) expression by miRNAs in HEK293T cells. Results of mRNA or protein levels are represented relative to negative control (Scrb). Each dot represents mean value from one independent experimental plate with 2–4 wells per treatment group. *—*p*-value < 0.05, **—*p*-value < 0.01, ***—*p*-value < 0.001.

**Figure 5 ijms-22-09691-f005:**
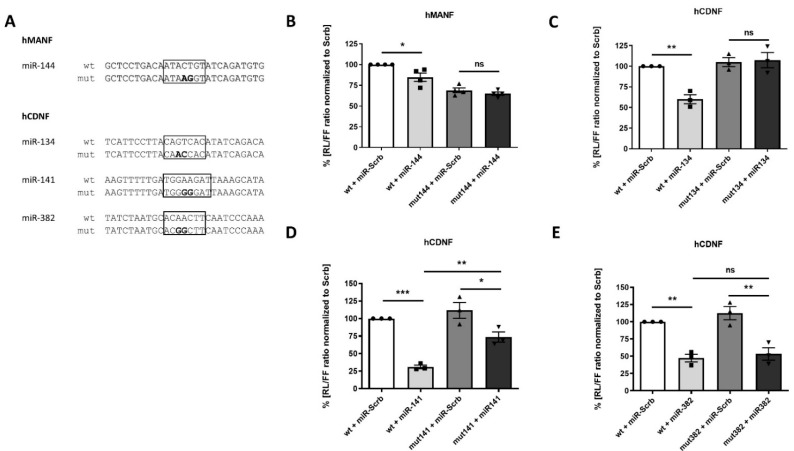
Regulation of luciferase activity via miRNA predicted binding sites. (**A**) Sequences of predicted miRNA binding sites: wild-type (wt) and mutated (mut). Predicted seed regions are framed. Mutations within binding sites of (**B**) miR-144, (**C**) miR-134, partially (**D**) miR-141, but not (**E**) miR-382, ablated ability of corresponding miRNAs to regulate luciferase signal. Each dot represents mean value from one plate with 4 wells per group. *—*p*-value < 0.05, **—*p*-value < 0.01, ***—*p*-value < 0.001.

**Figure 6 ijms-22-09691-f006:**
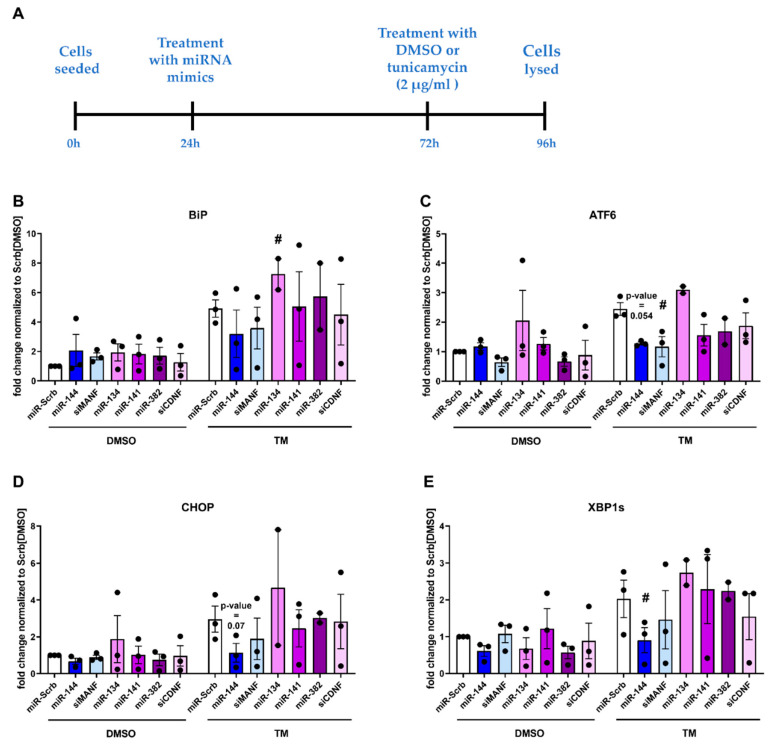
Expression of ER stress markers in HEK293T cells after treatment with miRNAs and DMSO or 2 µg/mL tunicamycin (TM) for 24 h. (**A**) Experiment timeline; Relative expression of BiP (**B**), ATF6 (**C**), CHOP (**D**) and XBP1s (**E**) mRNA. Each dot represents independent biological repeat. **#**—*p*-value < 0.05.

## Data Availability

The data that support the findings of this study are available from the corresponding author upon reasonable request.

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
