# Peer review of "Human-Specific Regulation of Neurotrophic Factors MANF and CDNF by microRNAs"

_ijms, 2021, doi:10.3390/ijms22189691_

Round 1

Reviewer 1 Report

In the present  article by Konovalova et al. entitled “Human Specific regulation of neurotrophic factors MANF and CDNF by MicroRNAs,” the author has predicted miRNAs for MANF and CDNF by using bioinformatics tools. Further authors have validated candidate miRNAs using the Dual-Luciferase reporter system and analysis of endogenous mRNA and protein expression level. However, the data looks weak and statistically less significant. The author further attempts to validate biological significance by checking expression levels of ER stress markers, but it appears inconclusive or insufficient to infer anything. Based on the Luciferase assay, the author claims that these miRNAs are human-specific. Although this is an exciting finding, several  concerns need to be addressed before accepting for publication:

  1. Related to Figure 1: An author can add an experimental scheme/workflow used to predict miRNAs
  2. Related to figure 2:
  3. Author could use mouse-specific miRNA as a control in Figures 2C and D? What is the statistical significance of miR544a in fig 2C.
  4. schematics of Luciferase assay in figure 2 for better understanding of readers.It is not necessary but helpful for readers if the author can add a few lines explaining how luciferase assay works?
  5. Related to figure 3: As mentioned by the author, several factors affect the regulation of gene expression by miRNA. To check endogenous protein levels HEK293 cell line was used. Can the author check the effect of studied miRNAs in other human cell lines with varying expression levels of MANF and CDNF (for example, SH-SY5Y)? Also, it will be good to include a mouse cell line as a control. This study's use of mouse cell line will help strengthen the authors claim that studied miRNA are human-specific.
  6. Related to figure 5: To check the biological significance of identified miRNAs, the authors check the expression level of ER stress markers by qPCR. However, this is a good approach; data provided by the author is not very conclusive and needs further experiments-
  7. Figure 5 needs to have proper labeling for each panel and citation in the figure legends.
  8. It appears that there is no statistically significant effect on the expression of ER stress markers. Although there is a decrease in XBP1s expression with miR-144, it is not very significant.
  9. The author does not mention/citethe first panel in figure 5 (BiP) in the text.
  10. Though there appears to be decreasing trend for levels of ATF6, CHOP, it appears that siRNA control (siMANF and siCDNF) also do not have a statistically significant effect on the expression level of ER stress markers, making it very difficult to conclude anything from figure 5.
  11. The author does not see any effect of siRNA or miRNA on any of the markers checked and thus needs to put more effort into identifying the physiological effect of there miRNAs.
  12. Does the author check if used siRNAs depleted protein levels? Western blot would be helpful.
  13. Although there is no significant decrease in mRNA expression level, the author should check if protein levels of ER stress markers were decreased (as the author did in figure 3C and 3D or by western blot). Alternatively, Several experimental approaches and reporter assays have been developed to study ER stress response; for example, author can check IRE activation or ATF6 translocation by fluorescence microscopy. This would help to demonstrate the physiological effect of studied miRNAs in ER stress response. (author should look at ER stress markers activity rather than mRNA expression level)
  14. Line 217, page8 – Authors have mentioned that“Our data show that functional miRNAs are essential for survival of adult dopaminergenic neurons (48)”. However, there is no g

Author Response

In the present article by Konovalova et al. entitled “Human Specific regulation of neurotrophic factors MANF and CDNF by MicroRNAs,” the author has predicted miRNAs for MANF and CDNF by using bioinformatics tools. Further authors have validated candidate miRNAs using the Dual-Luciferase reporter system and analysis of endogenous mRNA and protein expression level. However, the data looks weak and statistically less significant. The author further attempts to validate biological significance by checking expression levels of ER stress markers, but it appears inconclusive or insufficient to infer anything. Based on the Luciferase assay, the author claims that these miRNAs are human-specific.  

Thank you for your positive evaluation of our study and helpful comments to make it better! In order to clear your doubts about having rather not very robust effect of miRNAs on gene expression, we would like to refer to Linsen et al. (2008) review [1]reporting that miRNAs usually have mild effect on gene expressionUsually miRNAs function as fine-tuners of their targets’ expression. Rare robust downregulation can be observed in case of multiple miRNA’s binding sites in the 3’UTR of its target, which is not the case for MANF and CDNF. Therefore, our results are following current knowledge from the field. 

As miRNAs have multiple targetsit is challenging to demonstrate that observed effect of miRNA treatment is carried out via one specific target. Furthermore, since miRNA regulation of MANF and CDNF is human-specific, we were bound to use cell lines, which are very resistant to multiple stresses, including knockdown of pro-survival genes, such as CDNF and MANF. As can be seen from our data on Fig. 5, knockdown of MANF and CDNF with siRNA (without any additional stressors) had no effect on expression of ER stress markers. Considering all these circumstances, we believe that we made our best to perform current study and demonstrate effect of miRNAs on gene expression and, in case of miR-144, the effect on selected ER stress markersCertainly, physiological effects of miRNAs targeting MANF and CDNF need to be studied further, which will be a scope of our follow up manuscript 

Although this is an exciting finding, several concerns need to be addressed before accepting for publication: 

Related to Figure 1: An author can add an experimental scheme/workflow used to predict miRNAs 

Thank you very much for your comment! We have added the experimental workflow as Figure 1. 

Related to figure 2: 

Author could use mouse-specific miRNA as a control in Figures 2C and D? What is the statistical significance of miR544a in fig 2C. 

To our knowledge, currently there are no miRNAs reported to directly regulate mouse MANF or CDNF expressionWe attempted to use miR-338 as a possible control, because it is the only evolutionary conserved miRNA, predicted to have binding site on 3’UTR of both mouse and human MANF (as we have shown in Figure S1)Howevertreatment with this miRNA had no effect on expression of luciferase. 

p value for miR-544a in Fig. 2C is 0.09. We added this value on the revised Figure 3D (former Figure 2C).  

schematics of Luciferase assay in figure 2 for better understanding of readers. It is not necessary but helpful for readers if the author can add a few lines explaining how luciferase assay works? 

Thank you for valuable comment. We added both scheme (Figure 3A) and short explanation of the method in the Results section of the revised manuscript (lines 107-114). 

Related to figure 3: As mentioned by the author, several factors affect the regulation of gene expression by miRNA. To check endogenous protein levels HEK293 cell line was used. Can the author check the effect of studied miRNAs in other human cell lines with varying expression levels of MANF and CDNF (for example, SH-SY5Y)? Also, it will be good to include a mouse cell line as a control. This study's use of mouse cell line will help strengthen the authors claim that studied miRNA are human-specific. 

Thank you for your comment. At the beginning of the current studywe performed a pilot experiment using SH-SY5Y cells and tested MANF and CDNF levels in this cell line with ELISA. Our experiments showed that HEK293T cells had higher expression levels of MANF and CDNF and their expression was less variable during cell passaging. Therefore, we concentrated our attention on HEK293T cells. Nevertheless, although we did not test ability of selected miRNAs to regulate expression of MANF and CDNF in other cell lines, we speculated in the Discussion (lines 314-324), that most probably our findings can be applied in other cell types since no alternative 3’UTRs were reported for human MANF and CDNF 

The miRNAs tested in this study showed no ability to regulate 3’UTR of mouse MANF and CDNF in luciferase reporter system (Figure 3D and E). As we have mentioned in the revised Discussion section (lines 264-267), we cannot exclude the ability of selected miRNAs to regulate expression of mouse MANF and CDNF indirectlyHowever, as the focus of current study is direct miRNA regulation of MANF and CDNF, we believe the experiments using mouse cell lines fall outside the scope of this manuscriptWe agree that further tests of the ability of selected miRNAs to downregulate MANF and CDNF in mouse cell lines will be interesting to address in the follow up studies. 

Related to figure 5: To check the biological significance of identified miRNAs, the authors check the expression level of ER stress markers by qPCR. However, this is a good approach; data provided by the author is not very conclusive and needs further experiments- 

There are multiple reasons why we decided to use qPCR to assess physiological effect of selected miRNAsFirst, qPCR is one of the most used and well-recognized method to study ER stress [2]. It also has a lot of advantages for our study, such as ability to test multiple markers (in contrast to reporter assay, which usually shows expression of only one proteinand screen various conditions. In addition, qPCR is more sensitive than Western blot. As we have mentioned in our reply to the comments above, the effect of single miRNA on its target is rather subtleTherefore, we utilized the most sensitive assayNevertheless, we agree that this method has its limitations, and physiological effect of selected miRNAs need to be further studiedThis issue is addressed in the revised Discussion section (lines 321-322).  

However, we would like to emphasize that, in our view, these experiments fall outside the scope of current study. We aimed to show that miRNAs regulate expression of MANF and CDNF and this effect is human-specificdemonstrating that animal models have limitations in studying miRNAs and that evolutionary conservation is not prerequisite for miRNA functionality, at least for some targetsDuring the years of research since MANF/CDNF discovery, no miRNAs have been demonstrated to regulate their expression; this may have been due to reliance on animal, rather than human, modelsWe did not aim to study in details the function of selected miRNAs. This is the topic for another article, considering difficulty to relate physiological effect of single miRNA to its specific target. Nevertheless, we are currently developing a new very sensitive ER stress reporter assay, which shows promising preliminary results for the miRNAs regulating CDNF/MANFHoweverthe assay has to be further validated and is still a work in progress; we, therefore, could not include that preliminary data in the current manuscript.  

Figure 5 needs to have proper labeling for each panel and citation in the figure legends. 

We have updated the labeling and figure legend, as requested. 

It appears that there is no statistically significant effect on the expression of ER stress markers. Although there is a decrease in XBP1s expression with miR-144, it is not very significant. 

As shown on the Figure 6E, the difference in expression of XBP1s after treatment with miR-144 is statistically significant. We would like to point out that our data represent a result of three independent true biological repeats and not replicate wells on the same plate; therefore, we are confident in the robustness of our data. In addition, there is a trend for downregulation of ATF6 and CHOP (Figure 6C and D) after treatment with miR-144. ATF6 expression was downregulated after treatment with siRNA targeting hMANF and the effect of miR-144 on this ER stress marker is very similar. We changed graphs to make symbols identifying statistical significance more visible. Thank you for your feedback! 

The author does not mention/cite the first panel in figure 5 (BiP) in the text. 

Thank you very much for pointing this out; we have referred to Figure 6B in line 196-197 of the revised manuscript. 

Though there appears to be decreasing trend for levels of ATF6, CHOP, it appears that siRNA control (siMANF and siCDNF) also do not have a statistically significant effect on the expression level of ER stress markers, making it very difficult to conclude anything from figure 5. 

Knockdown of human MANF with siRNA caused significant downregulation of ATF6 and this effect was very similar to the one observed after treatment with miR-144 (p-value = 0.054). However, there was no effect of siMANF on other markers like XPB1s and CHOP, where downregulation (XBP1s) or a trend for it (CHOP) was observed after treatment with miR-144One of the possible explanations could be that miR-144 has other targets, which could affect ER stress response, such as Nrf2This is the reason why it is so hard to study physiological effect of miRNAs via specific targets. Nevertheless, thank you for pointing this out. We have further emphasized these findings in the revised Discussion (lines 302-313).  

There was indeed no effect of siRNA targeting CDNF on the expression of ER stress markershowever, in our opinion this is an important data to show. Previous results on CDNF knockout have been performed in animal model, but not in human cells. We have addressed these points in the revised Discussion section (lines 299-301). 

The author does not see any effect of siRNA or miRNA on any of the markers checked and thus needs to put more effort into identifying the physiological effect of there miRNAs. 

As it was discussed previously, siMANF and miR-144 downregulate ER stress markers ATF6 and XBP1s, respectively. In addition, miR-144 treatment has a strong trend to downregulation of ATF6 and CHOP. We changed the graphs on Figure 6 (former Figure 5) to make it easier to see these results 

Does the author check if used siRNAs depleted protein levels? Western blot would be helpful. 

siRNAs were used as a positive control in all experiments, where endogenous MANF and CDNF were tested. We showed in Figure 4C and D, that siMANF and siCDNF efficiently decreased protein levels of endogenous MANF and CDNF, respectively. Analysis was performed with ELISA, which is more sensitive and quantitative assay than western blot.  

Although there is no significant decrease in mRNA expression level, the author should check if protein levels of ER stress markers were decreased (as the author did in figure 3C and 3D or by western blot). Alternatively, Several experimental approaches and reporter assays have been developed to study ER stress response; for example, author can check IRE activation or ATF6 translocation by fluorescence microscopy. This would help to demonstrate the physiological effect of studied miRNAs in ER stress response. (author should look at ER stress markers activity rather than mRNA expression level) 

We fully agree with the reviewer’s point. As we discussed in the response to question 6, unfortunately, the sensitivity of current reporters and immunoassays is not sufficiently high to detect the changes in ER stress markers caused by studied miRNAsTo address this issue, we are currently developing new ER stress reporters, which demonstrate promising preliminary results with several miRNAs tested in this studyHowever, this is still work in progress, and, as validation of the assay will require more time, this is not the scope of current study. Nevertheless, we discussed limitations of our results in the revised Discussion section (lines 321-322).  

Line 217, page 8 – Authors have mentioned that “Our data show that functional miRNAs are essential for survival of adult dopaminergenic neurons (48)”. However, there is no graphs 

That was a reference to our previous research: we clarified it in the revised sentence (lines 229-231). Thank you for pointing out this unclarity!  

References:  

  1. Linsen, S.E., B.B. Tops, and E. Cuppen,miRNAs: small changes, widespread effects.Cell Res, 2008. 18(12): p. 1157-9. 
  2. da Silva, D.C., et al.,Endoplasmic reticulum stress signaling in cancer and neurodegenerative disorders: Tools and strategies to understand its complexity.Pharmacol Res, 2020. 155: p. 104702.

Reviewer 2 Report

The authors describe for the first time the possible role of miRNAs in the regulation of MANF and CDNF growth factor using in vitro assay. They determine the subpopulation of miRNA as potential modulator of MANF and CDNF expression by bioinformation analysis. Then these results were corroborate using Luciferase assay, qPCR and ELISA analysis. The authors identified that miR-144 controls MANF expression, and miR-134 and miR-141 downregulate CDNF levels.
Regards the physiological effect of this miRNA the authors explore the possible impact of miR-144, miR-134, miRNA 141 and miR-382 on Unfolded protein response, where they didn’t observe a clear effect in this process. 

Major comments
This investigation is a preliminary study of the impact of specific miRNAs on MANF and CDNF expression. The authors must be including stronger evidence about the relationship between these specific miRNAs on MANF and CDNF expression. 
The authors mention of several pathologies show alteration of MANF and CDNF levels. This alteration is modulated by increase of expression of these miRNAs. 
The authors must be exploring the possible mechanism involve in the regulation of this miRNA. 
The authors could determine or suggest the signaling or stimulus that trigger the regulation of this miRNA. 
The authors must suggest the physiological or physiopathology condition occur this phenomenon.

Author Response

The authors describe for the first time the possible role of miRNAs in the regulation of MANF and CDNF growth factor using in vitro assay. They determine the subpopulation of miRNA as potential modulator of MANF and CDNF expression by bioinformation analysis. Then these results were corroborate using Luciferase assay, qPCR and ELISA analysis. The authors identified that miR-144 controls MANF expression, and miR-134 and miR-141 downregulate CDNF levels. 

 Regards the physiological effect of this miRNA the authors explore the possible impact of miR-144, miR-134, miRNA 141 and miR-382 on Unfolded protein response, where they didn’t observe a clear effect in this process. 

Major comments 

This investigation is a preliminary study of the impact of specific miRNAs on MANF and CDNF expression. The authors must be including stronger evidence about the relationship between these specific miRNAs on MANF and CDNF expression. 

Thank you for your comment. In our study we demonstrated the ability of selected miRNAs to regulate 3’UTR of human MANF and CDNF using luciferase reporter assay, and, additionally, we tested their ability to regulate expression of endogenous MANF and CDNF on mRNA and protein levels using qPCR and ELISA, respectively. In addition, using reporter assay, we confirmed miRNA-mRNA binding sites for selected miRNAs. We strongly believe that accumulated data are sufficient to confirm conclusions of the current study. 

The authors mention of several pathologies show alteration of MANF and CDNF levels. This alteration is modulated by increase of expression of these miRNAs. 

This is a very good question which is difficult to address experimentally. Most of the data on the effect of altered levels of MANF and CDNF is observed in animal or in vitro models with deletion/overexpression of MANF/CDNF and inferred from these models [3-9]. Only few reports describe contribution of MANF on development of neurological conditions and nonautoimmune diabetes in patients with mutated MANF gene [3, 10]. There are several reports, showing alterations in levels of circulating MANF/CDNF in patients with diabetes [11], autoimmune diseases (rheumatoid arthritis and systemic lupus erythematosus) [12], PD [13] and stroke [14]. In addition, elevated levels of CDNF, but not MANF, were reported in the hippocampi of PD patients [15]. Altered miRNA profiling were reported in these diseases. However, it is impossible to make any conclusions about interplay between miRNAs and MANF/CDNF expression in patients with currently available data. In any case, this is exactly the question we are trying to rise with our study. 

The authors must be exploring the possible mechanism involve in the regulation of this miRNA. 

The authors could determine or suggest the signaling or stimulus that trigger the regulation of this miRNA. 

The authors must suggest the physiological or physiopathology condition occur this phenomenon. 

We thank the reviewer for these questions. Changes in miRNAs levels were detected in aging and multiple pathological conditions. For example, predominant downregulation of miRNAs was reported in brains of aged mice [16], in motor neurons of ALS (amyotrophic lateral sclerosis) patients and in cells, exposed to ER and oxidative stress [17]. In addition, our study [18] showed that there is downregulation of Dicer expression in dopamine neurons of aged mice, accompanied by general decrease in miRNAs expression. There are also data demonstrating reduced Dicer levels in dopamine neurons of PD patients [19, 20] as well as alterations in regulatory miRNA-mRNA networks in Parkinson’s disease [21] 

Many degenerative diseases, such as diabetes and Parkinson’s disease, are characterized by elevated ER stress. miRNA expression was reported to regulate and be regulated by ER signaling (reviewed in [22]). ER stress modulates expression of many miRNAs, e.g. via suppressing their biogenesis by translocation of Ago and Dicer proteins in stress granules or reducing expression of these proteins [17, 23]. However, there is also a fraction of miRNAs, whose expression is upregulated under stress conditions [16, 18]. In addition, multiple miRNAs were reported to affect expression of ER signaling members [22] 

These results suggest that when cells are struggling with ER stress, e.g. due to pathological protein accumulation in Parkinson’s disease, it affects miRNA biogenesis. Dysregulation in miRNAs expression further exacerbate ER stress, e.g. by affecting levels of UPR-involved proteins, making cells more vulnerable to additional stressors and further affecting miRNAs biogenesis, forming vicious cycle eventually contributing to cell death. However, to the best of our knowledge, there is no data on the molecular mechanisms regulating the expression of miRNAs studied in our manuscript; in fact, factors regulating expression levels of most miRNAs are not studied. Considering our results on the effects of miRNAs targeting MANF/CDNF, particularly miR-144, on ER stress markers, it would definitely be very informative to study molecular pathways and factors regulating miR-144 in stress conditions, which is the subject of our follow up studies. We have added these points in the revised Discussion section (lines 226-235, 277-287, 294-299). 

 References:  

  1. Linsen, S.E., B.B. Tops, and E. Cuppen,miRNAs: small changes, widespread effects.Cell Res, 2008. 18(12): p. 1157-9. 
  2. da Silva, D.C., et al.,Endoplasmic reticulum stress signaling in cancer and neurodegenerative disorders: Tools and strategies to understand its complexity.Pharmacol Res, 2020. 155: p. 104702. 
  3. Montaser, H., et al.,Loss of MANF Causes Childhood-Onset Syndromic Diabetes Due to Increased Endoplasmic Reticulum Stress.Diabetes, 2021. 70(4): p. 1006-1018. 
  4. Pakarinen, E., et al.,MANF Ablation Causes Prolonged Activation of the UPR without Neurodegeneration in the Mouse Midbrain Dopamine System.eNeuro, 2020. 7(1). 
  5. Tseng, K.Y., et al.,MANF Is Essential for Neurite Extension and Neuronal Migration in the Developing Cortex.eNeuro, 2017. 4(5). 
  6. Xu, S., et al.,Mesencephalic astrocyte-derived neurotrophic factor (MANF) protects against Abeta toxicity via attenuating Abeta-induced endoplasmic reticulum stress.J Neuroinflammation, 2019. 16(1): p. 35. 
  7. Lindahl, M., et al.,MANF is indispensable for the proliferation and survival of pancreatic beta cells.Cell Rep, 2014. 7(2): p. 366-375. 
  8. Zhou, W., et al.,Cerebral dopamine neurotrophic factor alleviates Abeta25-35-induced endoplasmic reticulum stress and early synaptotoxicity in rat hippocampal cells.Neurosci Lett, 2016. 633: p. 40-46. 
  9. Lindahl, M., et al.,Cerebral dopamine neurotrophic factor-deficiency leads to degeneration of enteric neurons and altered brain dopamine neuronal function in mice.Neurobiol Dis, 2020. 134: p. 104696. 
  10. Yavarna, T., et al.,High diagnostic yield of clinical exome sequencing in Middle Eastern patients with Mendelian disorders.Hum Genet, 2015. 134(9): p. 967-80. 
  11. Wu, T., et al.,Circulating mesencephalic astrocyte-derived neurotrophic factor is increased in newly diagnosed prediabetic and diabetic patients, and is associated with insulin resistance.Endocr J, 2017. 64(4): p. 403-410. 
  12. Chen, L., et al.,Mesencephalic astrocyte-derived neurotrophic factor is involved in inflammation by negatively regulating the NF-kappaB pathway.Sci Rep, 2015. 5: p. 8133. 
  13. Galli, E., et al.,Increased Serum Levels of Mesencephalic Astrocyte-Derived Neurotrophic Factor in Subjects With Parkinson's Disease.Front Neurosci, 2019. 13: p. 929. 
  14. Joshi, H., et al.,Decreased Expression of Cerebral Dopamine Neurotrophic Factor in Platelets of Stroke Patients.J Stroke Cerebrovasc Dis, 2020. 29(1): p. 104502. 
  15. Virachit, S., et al.,Levels of glial cell line-derived neurotrophic factor are decreased, but fibroblast growth factor 2 and cerebral dopamine neurotrophic factor are increased in the hippocampus in Parkinson's disease.Brain Pathol, 2019. 29(6): p. 813-825. 
  16. Inukai, S. and F. Slack,MicroRNAs and the genetic network in aging.J Mol Biol, 2013. 425(19): p. 3601-8. 
  17. Emde, A., et al.,Dysregulated miRNA biogenesis downstream of cellular stress and ALS-causing mutations: a new mechanism for ALS.EMBO J, 2015. 34(21): p. 2633-51. 
  18. Chmielarz, P., et al.,Dicer and microRNAs protect adult dopamine neurons.Cell Death Dis, 2017. 8(5): p. e2813. 
  19. Simunovic, F., et al.,Evidence for gender-specific transcriptional profiles of nigral dopamine neurons in Parkinson disease.PLoS One, 2010. 5(1): p. e8856. 
  20. Sutherland, G.T., et al.,A cross-study transcriptional analysis of Parkinson's disease.PLoS One, 2009. 4(3): p. e4955. 
  21. Santos-Lobato, B.L., A.F. Vidal, and A. Ribeiro-Dos-Santos,Regulatory miRNA-mRNA Networks in Parkinson's Disease.Cells, 2021. 10(6). 
  22. Maurel, M. and E. Chevet,Endoplasmic reticulum stress signaling: the microRNA connection.Am J Physiol Cell Physiol, 2013. 304(12): p. C1117-26. 
  23. Leung, A.K., J.M. Calabrese, and P.A. Sharp,Quantitative analysis of Argonaute protein reveals microRNA-dependent localization to stress granules.Proc Natl Acad Sci U S A, 2006. 103(48): p. 18125-30. 

Round 2

Reviewer 1 Report

The authors have addressed the comments properly. I would suggest the acceptance. 

Author Response

Thank you very much for positive evaluation of our manuscript.

This manuscript is a resubmission of an earlier submission. The following is a list of the peer review reports and author responses from that submission.

Round 1

Reviewer 1 Report

This manuscript by Julia Konovalova and colleagues reports on regulatory miRNAs modulating the expression of neurotrophic factors MANF and CDNF. In brief, human MANF is targeted by miR-144 and human CDNF is targeted by miR-134. In addition, these miRNAs could modify the expression of mRNA and protein of the neurotrophic factors.

In my view, this manuscript shows just an initial part of whole experiments. I think the approach is quite common, therefore not so much valuable information is included in the manuscript. The authors should show how miRNAs could affect the function of neurotrophic factors as well as why the finding of these miRNAs is important, at least.

Plus, I have serious concerns with this manuscript as indicated below.

  1. The authors used human embryonic kidney cell line for the experiments investigating neurotrophic factors. The authors need to justify why they use kidney cells, not neuronal or glial cells. Otherwise, they should use neuronal cells, glial cells, iPSC cells or primary culture.
  2. They showed the predicted target site of miRNAs on human neurotrophic factors in Fig.1. However, the results include mouse neurotrophic factors as shown in Fig. 2c and d. The result of miRNAs target site prediction for mouse neurotrophic factors are also needed.
  3. The effect of miR-144 on hMANF expressions seems very little in Fig. 3a and c. The authors need to discuss whether this kind of little change affects the function of neuronal or glial cells.
  4. For the effect of miR-134 on hCDNF mRNA, there is a tendency of downregulation but not statistically significant. If the authors insist this effect have some meaning, why not try to calculate the effect size?
  5. For the miRNA studies, antagomir is extremely useful tool. In this manuscript, the neurotrophic factors are downregulated by miRNAs so that antagomir might upregulate neurotrophic factors and possibly have a therapeutic potential for neurodegenerative disease.

In the present state, this manuscript is inappropriate for publication in the International Journal of Molecular Science.